# Sexual Violence against Children and Adolescents Taking Place in Schools: An Integrative Review

**DOI:** 10.3390/children7120258

**Published:** 2020-11-26

**Authors:** Charlene de Oliveira Pereira, Renata Macedo Martins Pimentel, Francisco Naildo Cardoso Leitão, Sandra Dircinha Texeira de Araújo Moraes, Paula Christianne Gomes Gouveia Souto Maia, Everson Vagner de Lucena Santos, Maria Nathallya Rodrigues de Freitas, Gildenia Pinto dos Santos Trigueiro, Petronio Souto Gouveia Filho, Luiz Carlos de Abreu

**Affiliations:** 1Laboratory of Study Design and Scientific Writing (LaDEEC), The ABC University Health Center (FMABC), Santo André 09060-590, Brazil; charlenepereira.pb@gmail.com (C.d.O.P.); re_pimentel1@hotmail.com (R.M.M.P.); nacal@outlook.com (F.N.C.L.); sandradircinha@gmail.com (S.D.T.d.A.M.); pcggsm@gmail.com (P.C.G.G.S.M.); eversonvls@hotmail.com (E.V.d.L.S.); nathallya.rodrigues@yahoo.com.br (M.N.R.d.F.); gil.trigueiro@gmail.com (G.P.d.S.T.); petronio_filho@yahoo.com.br (P.S.G.F.); 2Department of Integrated Health Education, Federal University of Espírito Santo (UFES), Vitória 29040-090, Brazil

**Keywords:** sexual violence, children’s health, adolescents’ health, school

## Abstract

Sexual violence against children and adolescents is considered a public health problem of worldwide scope. The objective is to analyze the findings in the literature that address the phenomenon of sexual violence against children and adolescents in the school environment; is an integrative literature review that has as its element-problem of interest children and adolescent victims of sexual violence in schools. The research filters used were: the availability of the text (free full text) and publication date (in the last five years); Initially, 1199 studies were identified, then, after application of filters and the removal of duplicated studies, a total of 175 studies was reached. Subsequently, the selection of articles occurred with the sieve of the titles, resulting in 20 studies. From these, 13 proceeded to the eligibility stage, with three being excluded after a full reading. Therefore, nine studies composed the final sample. One of the factors related to the occurrence of sexual violence against children and adolescents in schools is the absence of educational spaces on the subject, being the main parameter for approaching the outline of identification and prevention strategies, with the insertion of sexual education on the school routine, with the participation of the family.

## 1. Introduction

Sexual violence against children and adolescents is currently considered a public health problem of worldwide scope. It passes through different segments and social environments and the school is a common space of occurrence. The impacts of this type of violence are related to the psychosocial development and children’s and adolescents’ school performance, as well as their family members’ quality of life.

In definition, sexual violence is any action related to the use of physical force, coercion, intimidation or psychological influence to compel a person to have, witness or participate in sexual interactions for the purpose of profit, revenge or any other intention [1]. It can also be classified as sexual abuse and commercial sexual exploitation; intra-and extra-family sexual abuse; sexual exploitation in prostitution, pornography, sex tourism and trafficking in persons for sexual purposes [2]. From a chronological perspective, children and adolescents constitute the population between zero to nine and ten to 19 years of age, respectively [3].

The effects of sexual violence are complex and cover different consequences for the victim: physical problems, such as trauma, sexually transmitted infections; emotional problems such as phobias, anxiety, depression; behavioral, reaching social withdrawal or even inappropriate sexual behavior; in addition to cognitive distortions, such as self-blame [1,4,5].

The implications of sexual violence can also compromise the victim’s family dynamics. The emotional expressions the family members can experience managing the problem involve panic, anger, depression, crying, difficulties in establishing limits, and fear that the child reproduces violent relationships [4,6].

According to the United Nations (UN), violence against children and adolescents is still an ordinary silenced phenomenon, triggering a kind of scarcity of statistical data on the subject. Thus, all information and knowledge production about the characteristics of the event, victims and aggressors are essential for measures to be taken in order to reduce cases [7].

Sexual violence is an expression of gender violence and is considered a serious violation of women’s human rights [8]. Studies on sexual violence against children and adolescents, conducted in the Northeastern part of Brazil, identified a majority of female victims (73–79%) [9]. The results of the National School Health Survey (PeNSE), in its 2015 edition, identified: 4.0% of the students interviewed stated they were forced to have sexual intercourse, varying from 4.5% of girls to 3.7% of boys [10].

Currently, although there are Brazilian studies, there are still few records in the literature that specify the characteristics of the occurrence of sexual violence against children and adolescents in the school environment [11]. Highlight to the results observed in the study on the prevalence of sexual violence and associated factors among elementary school students, showing that sexual violence was more frequent among students who felt insecure on the school-to-school route home, at the school itself, as well as those who had been bullied [12].

Considering that sexual violence against children and adolescents in schools has represented a relevant and current issue for the health sector, either as an object of surveillance and monitoring or even as a priority focus of preventive interventions, the objective is to analyze the findings in the literature that addresses the phenomenon of sexual violence against children and adolescents occurred in schools.

## 2. Materials and Methods

This is an integrative literature review, a modality of retrospective and secondary study, whose objective is to describe, record, analyze and interpret data from previous studies on a given topic. As guidance for the preparation and planning of the research, the following steps were used: (1) formulation of the problem; (2) search in the literature; (3) pre-selection references and selection of included studies; (4) data extraction; (5) scientific data analysis and synthesis; and (6) interpretation of the results [13].

The formulation of the research problem—What are the parameters for approaching sexual violence against children and adolescents in schools?—Was outlined through the PICO model recommended to structure the research question construction, translated as an acronym for the elements population or problem, intervention or exposure, comparison or control, outcomes or results [14]. Thus, the population and exposure elements of this study comprise children and adolescents who are victims of sexual violence occurred in schools; and the result is the parameters for addressing this problem.

The literature search strategy was developed following these steps: determination of Descriptors in Health Sciences (Descritores em Ciências da Saúde—DeCS) in English sex offenses, child health, adolescent health and Schools; systematized searches in the electronic databases PubMed, EBSCOhost and LILACS, from January through February 2020. The research filters used were: availability of the text (free full text) and publication date (in the last five years).

The pre-selection of references and selection of included studies were conducted by two examiners who independently sought potentially eligible studies through the reading of the title and abstract, highlighting the articles that focused on the object of this study, namely: to characterize sexual violence against children and adolescents occurring in the school environment.

The process of searching, identifying, and selecting articles can be seen in Figure 1. Asa resulting, it identified 1199 (one thousand one hundred and ninety-nine) studies in the PubMed, EBSCOhost and LILACS databases, after the insertion of the DeCS ‘sex offenses’ and ‘child health’ and ‘adolescent health’ and ‘school’. Also, in the identification step, after the application of the filters ‘Full text available’ and ‘publication period between 2015 and 2019’ and the removal of duplicated studies, a total of 175 studies was reached.

Subsequently, the selection of articles occurred, first, with the sieve of the titles, resulting in 20 (twenty) selected studies, after the exclusion of 157 (one hundred and fifty-seven) that did not have the DeCS in their description. Subsequently, the screening through reading the abstracts gave 12 studies, seven being excluded because they did not specify the school as a place of the research and/or occurrence of sexual violence, as well as did not mention the age group of the victims.

From the 13 studies selected for the eligibility stage, four were excluded after a full reading since they did not address the research question—How is the approach of sexual violence against children and adolescents that takes place at school? The final sample was thus composed of nine studies.

## 3. Results

Through the complete reading of the studies included in the final sample, it was possible to produce a characterization of the selected articles, according to the author, year of publication, place where the study was developed, its objective and typology of the study. Results are shown in Table 1.

In Table 2 the result of the studies’ main findings synthesis can be observed, with the evidence specifications directed to the approach of sexual violence occurred in schools, in child and adolescent age groups.

## 4. Discussion

The evidence on the parameters for approaching sexual violence occurred in schools points to an outlined identification and prevention strategies, whose main intervention axis is the inclusion of sexual education in the school routine, with the participation of the family. They also signal the factors related to violence situations, the impacts on both children and adolescents’ mental health, highlighting the most common profile of exposure to the phenomenon.

The scope of intervention strategies for situations of sexual violence that have occurred in schools has diversified in the last years, moving through the implementation of prevention against violence, sexual education and reproductive health programs, also covering the report devices [20].

Studies [12,20,22] developed with the objective of evaluating the frequency and factors associated with situations of sexual violence against children and adolescents in elementary and high school in the United States, Ethiopia and Paraguay affirmed the increasing severity of the phenomenon and highlighted that one of the main factors related to this evidence is the absence of spaces for conversations and guidance on sexuality. They systematized the idea that, if efforts to introduce sex education are made early in the school and family environment, there may be a decrease in the risks of sexual violence incidence throughout individuals’ development.

Converging with the above evidence, it emphasizes the importance of using protocols as an innovative tool for individual and collective sexual education dissemination. In other words, a guide of how conducting the necessary information on institutional and cultural means for the defense against episodes of aggression, and the active participation of children and adolescents contributes significantly to schematize intervention models [21].

The relationships between the occurrence of bullying and sexual violence itself and the chances of being exposed to these situations, either as victims or as authors, have been consolidated as results highlighted in longitudinal studies developed with high school and university students in Canada and the United States, respectively. These outcomes evidence the spread of a cycle of violence, a path between indiscriminate violence and sexual violence [16,17].

Another longitudinal study conducted in Austin (TX, USA) aimed to examine the relationship between alcohol use and sexual abuse during childhood, on the formation of new victims and aggressors in the university environment. It was observed that young people who had a history of ‘blackouts’ due to excessive alcohol consumption had a higher chance of going through situations of exposure to sexual violence, when compared to young people who did not go through similar situations [18].

In this context, the importance of the family assumes marked consideration on the different studies reviewed. It is pointed out as a protective factor whenever they offer dialogue about sexuality on the daily household and participation in the school routine according to research conducted by Hebert et al. [16] and Mekuria, Nugussie and Abera [20].

The impacts of sexual violence on children and adolescents’ mental health were the subject of the Canadian study by Hebert et al. [16], whose objective was to explore the links between the types of abuse, the occurrence of bullying and mental health disorders. Their results showed that victims of sexual abuse had higher levels of psychological distress, low self-esteem, and suicidal ideation; the accumulation of traumatic events has been recognized as a risk factor for the symptoms production. However, we emphasize that the impacts are not linearly predictable and neither expressed solely through symptoms, since the relationships between the phenomena can encompass several potentializing factors.

Regarding reporting resources, Barr et al. [21] evaluated two types of methods (face-to-face-interview and sealed-envelope method) to increase the effectiveness of revelations of sexual abuse cases against children of a Ugandan primary school. Ultimately, impersonality and confidentiality were among the attributes pointed out as important characteristics for the effectiveness of this intervention modality.

In Brazil, female adolescents with black and brown skin and who live in conflicting family relationships make up the most common profile of exposure to sexual violence [10]. Converging with this panorama, Tordon et al. [19] studied the experiences of sexual abuse and mental health in a group of high school students in Sweden and stated that young people are exposed to risk situations, which, directly or indirectly, can influence their mental health. They also remarked the urgent demand to establish effective case tracking, their follow-up and resolution.

A survey by [13] conducted from 2010 to 2014, in the school environment, throughout Brazil, found 2226 reports of sexual violence against children and adolescents within schools. The profile of the majority of the victims was female (63.8%) with white skin (51.8%). Among the types of violence evidenced were rape (60.9%), sexual harassment (29.7%) and indecent assault (21.6%), with a reported recidivism rate in about one-third of cases (34.7%). Regarding the profile of the aggressor, the majority were male (88.9%) and known by the victim (46%).

In a study conducted with 6709 students from public schools in ten Brazilian capitals, it was found that 1.6% of the adolescents interviewed reported having suffered sexual violence within the school, and 5.6% reported knowing that sexual violence occurred in their surroundings. This panorama corroborates the discourse on and the feeling that schools are no longer a safe place for students, previously considered a place deserving of care and respect [15].

However, even facing the weaknesses established by the school institution itself, it constitutes a strategic locus for monitoring the protective and risk factors of children and adolescents. In particular, due to its characteristics as an interactions, experiences, and socialization provider, composing one of the extreme influential environments on the individual’s cognitive, social and emotional formation [23].

In Brazil, the Federal Government has implemented the Projeto Escola que Protege (School That Protects) project since 2004, in order to expand the dialogue between school and society, in searching for articulation means that allow greater protection of children and adolescents against different types of violence, developing then integrating actions between health and education, with essential families’ participation [2,24].

A field research on violence in schools in seven Brazilian capitals conducted with elementary and high school students highlighted the role of teachers as knowledge mediators on the subject. From the students’ perspective, violent attitudes are often related to the school structure, the neglect of teachers and other school staff and the lack of discussion on the subject [25].

Therefore, one of schools and educational institutions’ main responsibilities is to provide a safe environment to full learning and healthy development of children and adolescents, protecting them from risk situations of physical and mental health.

The limitations found in the present study are specific to research with primary sources of information from electronic bibliographic databases, the use of descriptors and filters may have resulted in the exclusion of important reports on the subject.

## 5. Conclusions

The literature findings highlight that one of the factors related to the occurrence of sexual violence against children and adolescents in schools is the absence of educational spaces on the subject, being the main parameter for approaching the outline of identification and prevention strategies, capable of ensuring the insertion of sexual education in the school routine, with the participation of the family.

Corroborating the relevance of the theme, the present study contributes with some parameters necessary to characterize the ways of approaching sexual violence against children and adolescents occurred in schools. They are: (1) What to do? Offer of educational activities; (2) To whom? Children and adolescents, involving family participation; (3) What’s the point? With the objective of informing and promoting reflection on sexual violence; (4) In what way? Individual and collective, using information and creative methodological and audiovisual resources; (5) What contents? Violence, sexuality, adolescents’ sexual and reproductive health, body care, use of alcohol and other drugs, relationships with classmates and family.

## Figures and Tables

**Figure 1 children-07-00258-f001:**
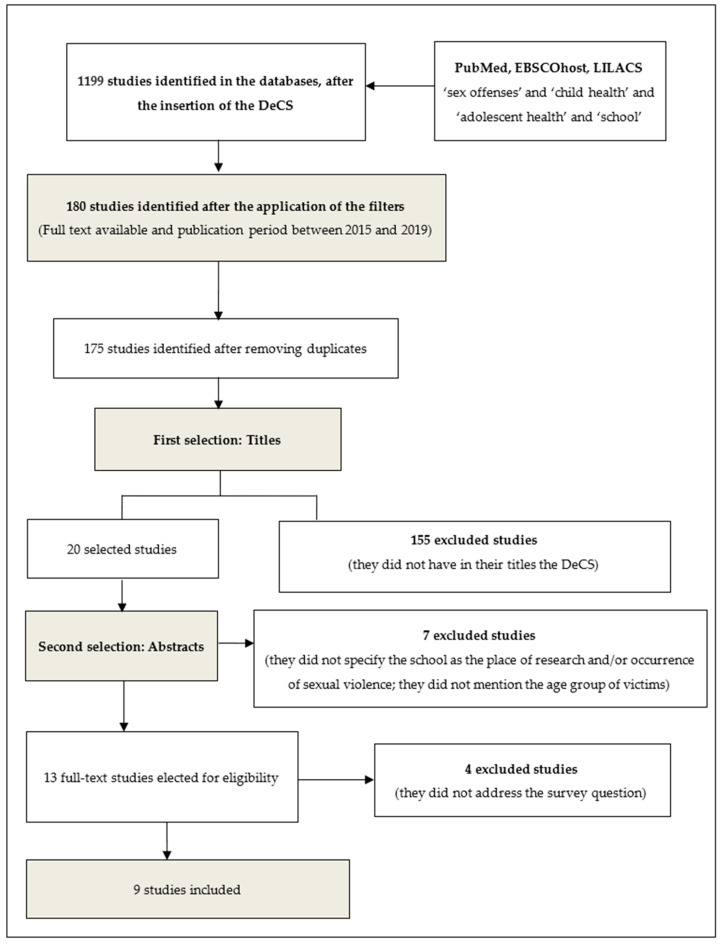
Flowchart of the studies included, with the process of searching, identifying, and selecting articles.

**Table 1 children-07-00258-t001:** Characterization and quality of selected studies (2015 to 2019).

Authors, [Reference] Year	Objective	Type of Study
Basile et al. [12] 2018	To analyze the protective factors against the occurrence of sexual violence in elementary and high school students.	Longitudinal/Observational
Santos et al. [15] 2019	To describe the prevalence of sexual violence among adolescent students and identify factors associated with this phenomenon.	Transversal/Observational
Hebert et al. [16] 2016	To explore the links between the types of sexual abuse and the occurrence of bullying and mental health disorders among high school students; and study maternal support as a potential protective factor.	Longitudinal/Observational
Espelage et al. [17] 2016	To describe the relationship between the bully and the possible evolution to sexual harassment and aggression among high school students.	Longitudinal/Observational
Wilhite, Mallard, Fromme [18] 2019	To examine the relationship between the use of alcohol and sexual abuse during childhood on the formation of new victims and aggressors in the university environment.	Longitudinal/Observational
Tordon et al. [19] 2018	To compare experiences of sexual abuse and mental health in a group of high school students living outside their homes, with a representative sample of classmates of the same age and similar schooling living with their parents.	Transversal/Observational
Mekuria, Nigussie, Abera [20] 2015	To assess the prevalence and factors associated with child sexual abuse in high school adolescent students in an African city.	Transversal/Observational
Barr et al. [21] 2017	To evaluate two types of specific methods and (face-to-face-interview and sealed-envelope method) to increase the reporting of cases of sexual abuse against children in a primary school.	Transversal/Observational
Suhurtl et al. [22] 2015	To determine the frequency of sexual abuse and other forms of violence in high school students.	Transversal/Observational

Source: Compiled by the authors, 2020.

**Table 2 children-07-00258-t002:** Synthesis of evidence to address sexual violence in schools, in the child and adolescent age groups (2015 to 2019).

Authors [Reference] Year	Evidence to Approach
**Children**	
Basile et al. [12] 2018	They systematized the idea that, if efforts to introduce sexual education and other strategies to prevent violence are made early, during adolescence and involving parents and schools, there may be a decrease on the incidence of sexual violence risk throughout their development of individuals.
Espelage et al. [17] 2016	They affirm a relationship between the practice of bullying at the beginning of high school and the practice of sexual harassment at the end of studies, which reveals a model on the path between indiscriminate violence and sexual violence.
Barr et al. [21] 2017	They point out attributes such as impersonality and confidentiality as important choices to characterize reporting methods for cases of child sexual abuse.
**Adolescents**	
Santos et al. [15] 2019	They show the most common profile of exposure to sexual violence, namely: young people aged around 13 and 16, female, black and brown skin, conflicted social relationships with classmates and family members. They also show that there is a great feeling of insecurity from the young people, even in the school environment, which should be taken as safe.
Hebert et al. [16] 2016	Based on the conclusion that maternal support can reduce the risk of developing mental health problems, the authors contribute to the outline of strategies for prevention, intervention and protection of young people exposed to situations of abuse, such as parental involvement.
Wilhite, Mallard, Fromme [18] 2018	They pointed out the relationship between alcohol use and the chance of being exposed to situations of sexual violence, either as the victim or as the author.
Tordon et al. [19] 2018	They concluded that, even with the educational resources provided by the institutions, young people are still exposed to risky situations, which directly or indirectly might influence their mental health. Thus, it is necessary to establish a routine, and effective tracking measures for the cases screening and their monitoring and resolution.
Mekuria, Nigussie, Abera [20] 2015	They proved that the levels of childhood sexual abuse against girls were very high. One of the main factors related to this evidence was the family environment conversation about reproductive health, showing that, in families in which this type of conversation happened, there was a lower rate of cases.
Suhurtl et al. [22] 2015	They affirmed the increasing severity of violence and highlighted children and adolescents’ vulnerability and the need to implement prevention programs toward violence and, especially, sexual abuse.

Source: Compiled by the authors, 2020.

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
