# Peer review of "Sexual Violence against Children and Adolescents Taking Place in Schools: An Integrative Review"

_children, 2020, doi:10.3390/children7120258_

Round 1

Reviewer 1 Report

The work is presented in a clear and fluid way both in the presentation of the contents and in the method used. I think it is possible to appreciate its publication.

Author Response

Dear reviewer 1,

Very respectfully, we wrote to thank you for your attention and the punctuated comments about our manuscript.

With our best wishes,

Charlene Pereira, Renata PimentelFrancisco Leitão, Paula Maia, Everson Santos,

Maria Nathállya de Freitas, Gildenia Trigueiro, Petronio Gouveia Filho and Luiz Abreu.

Reviewer 2 Report

The article deals with a current and socially pertinent topic. The study of sexual violence in schools is of great relevance for preventing this phenomenon.

Once the concept is presented, the authors define the age limits (0-9 and 10-19) and here the first question arises: what parameter do they use to define these limits? Age limits are not universal if we consider a country's legal codes or even international regulations such as the one it cites (WHO) or others (e.g., Convention on the Rights of the Child). Unlike European countries, Brazil is not a signatory to the Istambul Convention, but has other important legal references (e.g., Brazil's Statute of the Child and Adolescent). So it is suggested that the authors explain and argue the choice they made for these age limits.

Likewise, when referring to the consequences of sexual violence, it is important to be aware that the impact of this experience is not linearly predictable. There are asymptomatic children and young people, even victims, with evidence of signs and symptoms such as the ones they suggest. It is important to express this, base it on studies, and not create an idea that the recognition of the phenomenon is necessarily induced by the presence of these indicators.

The authors mention an important aspect, which is the implications for family dynamics. As it is an abusive experience outside the family, and given the role that the family can play in signaling and support, the authors may defend the importance of family being supported to be able to participate in the intervention.

Authors must support their statements in evidence or references (e.g., Currently, although there are Brazilian studies ...)

The authors state that they will carry out an integrative review, indicating the steps, but at a certain point mention the PICO system used in systematic reviews (SR). SRs require that a specific research protocol be followed (please see: doi: 10.1186/2046-4053-4-1) and require steps that this article does not mention (e.g., critical evaluation of articles; peer review of methodological validity). The evaluation of the quality of the articles must be included between the stages, either an integrative review or a systematic review, so it is suggested to present a complimentary table, even if they do not want to publish it.

Aware of the defined inclusion and exclusion criteria, authors are asked to help understand the following:

  1. because, in the 'language' criterion, Portuguese was not included (if the Brazilian context matters for the review)?
  2. the reason for the 5-year period (excluding studies on the topic, including in Brazil, in the period below 2015);
  3. why the term used in the title (sexual violence) is not among the search passwords?
  4. how many researchers were involved in the selection of articles and what evaluation procedure did they adopt?
  5. was the snowball process covered? (because the authors refer to «…the use of descriptors and filters may have resulted in the exclusion of important reports on the subjec

Review studies are essential for producing relevant social changes such as those proposed in this article. The authors are encouraged to make the methodological part of the review more robust, clarifying whether it is integrative or systematic, strictly following the steps and rules (e.g., the authors use the flow diagram of the systematic review protocol but have changed the schematic rules of informing about the exclusion of articles – see http://prisma-statement.org/prismastatement/flowdiagram.aspx).

The conclusion should be further improved. The contents presented at the end of the discussion on proposed themes for the intervention make more sense to be discussed at the conclusion but in a more improved way.

Author Response

Dear reviewer 2,

Very respectfully, we wrote to thank you for your attention and for all the punctuated comments about our manuscript.

All accepted recommendations and the reformulations listed in the manuscript, related to the flowchart and conclusions, are located in the new redesigned version on November 11, 2020, respectively: line 95, page 3; line 295, page 7.

In addition, we point out that the parameters used to define age limits to characterize the categories of children and adolescents were the definitions of the World Health Organization (WHO), as mentioned on page 2, lines 38 and 39, because the study intended to outline the approach of the phenomenon in other contexts, not just the Brazilian. In this sense, WHO consolidates itself as an international reference organization for the production of studies and interventions on violence against children and adolescents.

About the impact of the experience with the phenomenon, in fact, is not linearly predictable. The very concept and characteristics of the phenomenon point to this. Another excellent observation is about the importance of the family being supported in order to participate in the intervention. This is nuclear. Like schools, the absence of spaces for dialogue within families was one of the elements pointed out in the studies as aggravating the phenomenon.

Our study is characterized as an integrative review, as explained on page 2, line 65. The evaluation of the articles' quality included the observation of essential items that should be considered in the writing of observational studies, according to the statement Strengthenig the Reporting of Observational Studies in Epidemiology (STROBE). The complementary tables used are attached for purposes of assessment and not necessarily for publication.

Regarding the inclusion and exclusion criteria, we explain: In the “language” criterion, Portuguese was not included because we understand that the most robust publications on the subject, including Brazilian ones, would be included in the data repositories that cover the English language; The reason for the five-year period was aimed at capturing the evolution of the approach to the problem more recently, given the importance of the theme and the need for mothers to need daycare centers and schools to be effectively inserted into the labor market; The sex offense descriptor composed one of the search passwords, as described on page 2, line 79; In the selection of articles, four researchers were involved; The evaluation procedure used by the authors included the observation of essential items that should be described in observational studies, according to the statement Strengthenig the Reporting of Observational Studies in Epidemiology (STROBE); The snowball process was not used.

Our study is characterized as an integrative review, as explained on page 2, line 65. However, we recognize and agree that the flowchart describing the stages of the review needs to be modified in order to include the inclusion and exclusion criteria of the studies.

Finally, we would like to thank once again all the comments and contributions presented.

With our best wishes,

Charlene Pereira, Renata Pimentel, Francisco Leitão, Paula Maia, Everson Santos, Maria Nathállya de Freitas, Gildenia Trigueiro, Petronio Gouveia Filho and Luiz Abreu.

Reviewer 3 Report

An interesting article, although it requires some additional work, from the abstract to the conclusion.

See my notes below:  

The abstract requires reformulating, so that it acts as a summary of the entire article, and not just individual elements. I recommend removing the words '(1) Background:; "(2) Methods:” "(3) Results:” "Conclusions:”, and instead creating a more cohesive format.  

Line 52-55 I believe that since the article is not entirely based on the Brazilian context, the introduction should contain statistics from different countries as well, as well as comparisons. On top of this, statistics from five years ago are a little dated – there should at least be a reasoning as to why these stats were chosen.  

Line 71: Authors didn't clearly say what the research questions are.  

I believe that Figure 1 should be part of the text, and not added in the appendix, as this makes it more difficult to read. It would be best to add Figure 1 after line 85. These notes are related also to the Results. Table a1 and a2 should be included and analyzed in the text, and not added to the Appendix.  

I don't understand why the authors only reference Brazil – is this the purpose of the article? From the Materials and Methods and abstract, this doesn't seem to be the case, and since this isn't the purpose of the article, the authors should reference observations from a multicultural perspective. At the same time, the discussion is not well-supported by the results.

I have a similar note for the conclusion, which does not reference the analysis, and instead is a general conclusion. I recommend expanding the context of the analysis – I will note that the conclusion is not well-supported by the results, either.  

Anything included in the references should be translated into English and included in brackets.

I recommend moderate changes, as there are many errant commas scattered at inappropriate points in sentences throughout the text, as well as a misuse of semi-colons where commas or colons might be more appropriate. I have also marked sentences with larger grammatical errors in the PDF.

Author Response

Dear reviewer 3,

Very respectfully, we wrote to thank you for your attention and for all the punctuated comments about our manuscript.

All accepted recommendations and the reformulations listed in the manuscript, related to the abstract and the research question, related to figures and tables are located in the new redesigned version on November 11, 2020, respectively: lines 11, 15, 16, 20, page 1; line 71, page 2; lines 94, 161 and 166, pages 3,4 and 5.

In addition, we point out that the statistical data presented were chosen, even corresponding to five years ago, because they address the specificity of the object of study.

Finally, we would like to thank once again all the comments and contributions presented.

With our best wishes,

Charlene Pereira, Renata Pimentel, Francisco Leitão, Paula Maia, Everson Santos, Maria Nathállya de Freitas, Gildenia Trigueiro, Petronio Gouveia Filho and Luiz Abreu.

Round 2

Reviewer 2 Report

The authors answer most of the questions raised in the review, leaving only a few points that I think can be re-evaluated for inclusion.

  • It is important to express in the text the possibility of asymptomatic victims;
  • Authors write in the plural («Currently, although there are Brazilian studies ...) but only have one reference. Include at least one more
  • The flow chart does not reveal changes. For example, you cannot take 155 articles out of 20. The exclusion box must be before the selection

A single note: the question was why they did not include in the passwords for selection exactly the term "sexual violence" (it is not the same as sexual offenses) … but it is just to think, they do not need to answer.

Author Response

Cover Letter

Dear reviewers,

We would like to thank once again all the comments and contributions presented.

All accepted recommendations and the reformulations listed in the manuscript, related to the impact of the experience with the phenomenon, the flowchart and the Brazilian studies, are located in the new redesigned version on November 17, 2020, respectively: line 58, page 2; line 98 , page 3; line 209, page 6.

With our best wishes,

Charlene de Oliveira Pereira, Renata Macedo Martins Pimentel, Francisco Naildo Cardoso Leitão, Sandra Dircinha Texeira de Araújo Moraes, Paula Christianne Gomes Gouveia Souto Maia, Everson Vagner de Lucena Santos, Maria Nathállya Rodrigues de Freitas, Gildenia Pinto dos Santos Trigueiro, Petronio Souto Gouveia Filho and Luiz Carlos de Abreu.

Reviewer 3 Report

I accept all changes that have been made by the authors following my comments and suggestions. 

Author Response

Dear reviewer 3,

We would like to thank once again all the comments and contributions presented.

With our best wishes,

Charlene Pereira, Renata Pimentel, Francisco Leitão, Paula Maia, Everson Santos, Maria Nathállya de Freitas, Gildenia Trigueiro, Petronio Gouveia Filho Luiz Abreu